# Benchmarking Complex Chart Reasoning via Sub-question Decomposition and Variant-based Robustness Analysis

## Abstract

Multimodal Large Language Models (MLLMs) demonstrate strong potential for chart interpretation, yet existing benchmarks mainly assess final-answer accuracy, neglecting intermediate reasoning validity and robustness to visual perturbations. We present CHART-FGR, a fine-grained benchmark that decomposes each complex question into interpretable sub-questions and tests models under five visual perturbations (blur, noise, watermark, label removal, color distortion). Spanning 20 chart types, 200 base charts yield 1,652 sub-questions and 8,260 QA pairs across 1,000 images. Evaluations of leading MLLMs show significant performance drops (18–42 %) and reveal that most failures stem from early decomposition or perception errors. These findings highlight the necessity of process-oriented diagnostics to ensure trustworthy deployment in real-world, low-quality visual environments. Code is available at https://anonymous.4open.science/r/ChartSQA-DACC/.

## 1 Introduction

Multimodal Large Language Models (MLLMs) exhibit substantial potential for interpreting charts, a critical skill for applications in automated data analysis and report generation. Chart understanding task requires multi-step reasoning, where models are required to deconstruct problems, extract relevant information, and perform analysis. However, evaluating the advanced reasoning capabilities of these models remains a significant challenge.

Current chart understanding benchmarks suffer from a fundamental limitation: **they almost exclusively focus on the correctness of the final answer while overlooking the logical soundness of the intermediate reasoning process**. This evaluation paradigm is particularly inadequate for complex reasoning tasks, where a correct final answer can mask critical errors in intermediate steps, leading to a misleading assessment of a model's true capabilities. For example, as shown in Figure 1(a), although the model correctly answers "No" to the question about the sum of the two smallest bars, it misidentifies the category of the ninth largest bar, indicating a flaw in intermediate reasoning. Similarly, in Figure 1(b), the model incorrectly answers "4" instead of the correct "3" to a question requiring value extraction and conditional filtering, but current benchmarks do not identify whether the error is due to faulty extraction or logical filtering. These gaps prevent a clear diagnosis of whether failures arise from perceptual issues or reasoning flaws, resulting in misleading evaluations of model performance.

Additionally, **the performance of Multimodal Large Language Models (MLLMs) can be fragile, exhibiting sensitivity to visual perturbations commonly encountered in real-world scenarios**. Charts may suffer from low resolution, visual noise, or the presence of watermarks, yet current benchmarks fail to systematically evaluate model robustness against such degradations. This oversight hinders a comprehensive understanding of model reliability and resilience in practical applications.

To address these limitations, we propose a fine-grained benchmark for chart understanding, introducing two key innovations. First, we decompose complex reasoning tasks into logically coherent sub-questions, allowing us to evaluate whether a model's reasoning path aligns with intermediate steps and whether it can reach the correct final answer when guided by the correct path. This design helps

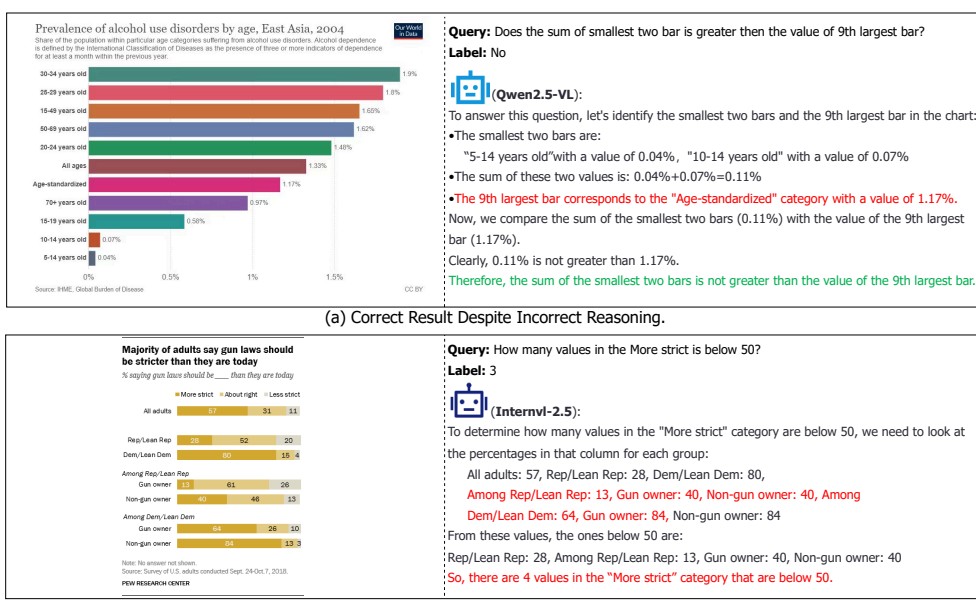

(a) Correct Result Despite Incorrect Reasoning.

(b) Incorrect Result and Unidentified Reasoning Error.

Figure 1: Shortcomings of existing evaluation methods in ChartQA—(a) Correct answer with incorrect reasoning; (b) Incorrect answer with undetected error. Green highlights correct conclusions; red highlights incorrect ones.

diagnose whether a model's failure lies in task decomposition or execution. Second, to assess model robustness, we generate multiple chart variants simulating real-world perturbations. Label-removed charts test reliance on structural and visual cues without textual labels. Noise- and watermark-added charts measure resilience to visual clutter, while blurred charts simulate low-resolution scenarios, evaluating global perception of shapes and spatial relationships. These variants enable us to analyze how different types of interference impact specific reasoning steps, providing insights to improve robustness in chart reasoning.

To support this framework, we construct a large-scale, fine-grained benchmark encompassing 20 different chart types, including bar, line, stacked, pie, and radar charts. In total, we generate 200 original charts, each paired with a complex reasoning question that is further decomposed into 4–10 sub-questions, resulting in approximately 1,652 question-answer pairs. Each chart is also rendered into five variants (original with labels, blurred, noise-added, watermark-added, and label-removed), leading to 1,000 chart images and a total of 8,260 question-answer pairs. Our primary contributions are threefold:

Multimodal Large Language Models (MLLMs) show strong potential for chart interpretation, yet existing benchmarks largely evaluate only the correctness of final answers, overlooking the validity of intermediate reasoning steps and robustness under visual perturbations. This limitation is particularly critical for complex reasoning tasks, where correct outcomes may mask faulty decompositions or flawed logic, and where real-world degradations such as blur, noise, or missing labels can severely impact performance. To address these gaps, we introduce a fine-grained benchmark for complex chart reasoning that combines sub-question decomposition with variant-based robustness analysis. Each complex question is systematically broken into logically coherent sub-questions, enabling interpretable evaluation of reasoning pathways and precise error diagnosis. In addition, we construct five variants of each chart—including blurred, noise-added, watermark-added, and label-removed versions—to measure model resilience under degraded conditions. Our benchmark covers 20 chart types with 200 original charts, yielding 1,652 sub-questions and 8,260 QA pairs across 1,000 chart images. Experiments on state-of-the-art MLLMs reveal distinct weaknesses in decomposition, perception, and robustness, underscoring the necessity of fine-grained evaluation for advancing reliable chart reasoning.

## 2 RELATED WORK

Chart understanding tasks require models to interpret both the visual and textual elements of charts and respond accurately to a variety of instructions. Several benchmark datasets have been proposed to evaluate the capabilities of Multimodal Large Language Models (MLLMs) in these tasks, including Chart Question Answering (CQA) (Masry et al. (2022); Methani et al. (2020); Kantharaj et al. (2022a)), chart summarization (Tang et al. (2023); Kantharaj et al. (2022b); Rahman et al. (2022)), chart-to-table conversion (Xia et al. (2023); Chen et al. (2024a)), and chart re-rendering (Moured et al. (2024); Yang et al. (2024)).

CQA has emerged as a core benchmark for assessing chart understanding. Early datasets like FigureQA (Kahou et al. (2017)) generated over a million binary (yes/no) questions using templates across synthetic chart types, while DVQA (Kafle et al. (2018)) expanded the template set and constrained answers to a 1,000-word vocabulary. Despite their scale, these datasets were limited in chart diversity and question complexity. Later efforts such as OpenCQA (Kantharaj et al. (2022a)) and ChartQA (Masry et al. (2022)) crowdsourced open-ended QA pairs from real-world charts, introducing greater linguistic variety and higher reasoning demands. More recent datasets, including ChartX (Xia et al. (2024)) and CharXiv (Wang et al. (2024)), further expanded chart types, domains, and visual complexity. These benchmarks primarily evaluate final-answer correctness using metrics like Exact Match (EM), Accuracy, and Average Normalized Levenshtein Similarity (ANLS). While these metrics measure output accuracy, they do not capture the reasoning process behind the answers. Metrics such as CHAIR (Rohrbach et al. (2018)) address issues like object hallucination but still focus on outputs rather than intermediate reasoning steps.

With the rapid advancement of reasoning models, modern MLLMs can perform multi-step reasoning on complex chart understanding tasks. However, no benchmark currently exists to systematically evaluate this capability. Building such a benchmark poses two main challenges: dataset construction and evaluation design. The first challenge is creating data and metrics that accurately assess the correctness of each reasoning step while accounting for dependencies and causal relationships—a crucial aspect, as existing benchmarks focus primarily on final-answer accuracy.

## 3 BENCHMARK

We first highlight two core principles that guided our benchmark design: **Stepwise Evaluation**, which decomposes complex queries into ordered sub-questions to trace a complete chain of reasoning. This enables assessment of intermediate reasoning steps—not just final answers—while capturing the dependencies and causal relationships that drive multi-step inference; and **Robustness through Visual Diversity**, which introduces controlled visual variants of each chart (e.g., blurred, noisy, watermarked, or with annotations removed) to probe model resilience to realistic visual perturbations, ensuring the benchmark evaluates both reasoning accuracy and stability under imperfect, real-world conditions.

To systematically evaluate the chart reasoning capabilities of multimodal large language models (MLLMs), we construct a hierarchical and controllable benchmark dataset. The construction process is designed to ensure diversity across chart types, define a clear hierarchy of tasks, and regulate reasoning complexity. As illustrated in Fig. 2, the dataset is built through a multi-step pipeline that integrates automatic generation via large language models with subsequent manual refinement. The main stages are as follows:

### 3.1 CHART IMAGE GENERATION

**Original Image Generation.** We selected 20 commonly used chart types, including single-series charts (such as line charts, pie charts, and rose charts) and multi-series charts (such as multi-line charts, stacked bar charts, and box plots).

For each chart type, we created a structured JSON template specifying elements such as chart type, title, topic, axis labels, series, values, and color schemes. Using these templates, GPT-4o generated datasets aligned with topics randomly chosen from 20 predefined domains (e.g., science, economics, climate). The data were then rendered into charts with Python libraries like Matplotlib and Plotly. We produced 50 images per chart type, yielding a total of 1,000 charts.

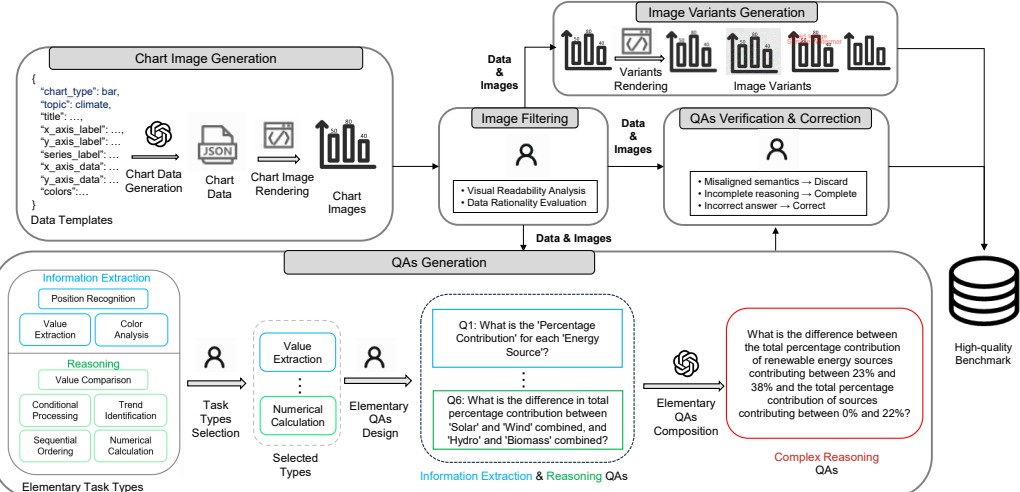

Figure 2: Dataset Construction Process.

**Chart Image Filtering.** To guarantee quality, we employed a two-step evaluation procedure:

*Step 1: Chart Scoring.* Each chart was evaluated along two dimensions: *visual readability* and *data rationality*.

- *Visual readability* assesses the clarity of key chart components, such as axis labels, legends, and data values. Charts received 2 points if all elements were clearly visible and unobstructed, 1 point if some elements were partially occluded but remained readable, and 0 points if elements were severely obstructed or blurred.

- *Data rationality* evaluates whether the chart adheres to logical and statistical norms. For instance, pie chart segments should sum to 100%, quartiles in box plots must follow the correct order, and radar chart axes need to be dimensionally consistent. Charts with fully consistent and logically correct data received 2 points, while any inconsistency or error led to a score of 0.

Each chart was independently scored by three human reviewers. For each reviewer, the total score per chart was calculated as the sum of the two dimensions, resulting in a score between 0 and 4. The final score for each chart was computed by averaging the scores from all three reviewers.

*Step 2: Chart Filtering.* After scoring, charts were ranked within each category based on their average scores. The top ten charts from each category were selected, resulting in a final set of 200 high-quality chart images, denoted as $\mathcal{I} = \{I_j\}_{j=1}^{200}$.

## 3.2 QUESTION–ANSWER PAIR GENERATION

We begin by defining a set of fine-grained task types for chart reasoning, organized into two categories. The first category, **Information Extraction**, focuses on directly retrieving visual elements from charts and includes: (1) *Value Extraction* – identifying specific data values, (2) *Color Identification* – determining the color associated with a chart element, and (3) *Position Recognition* – locating the position of visual items such as bars, points, or segments. The second category, **Reasoning**, requires interpreting and reasoning over the extracted information and includes: (1) *Value Comparison* – comparing numerical values between elements, (2) *Conditional Processing* – making decisions based on specific values or conditions, (3) *Trend Identification* – recognizing patterns such as increases, decreases, or stability, (4) *Sequential Ordering* – determining the rank or order of elements by value, and (5) *Numerical Calculation* – computing derived results such as sums, differences, or averages.

To evaluate multi-step reasoning over charts, we construct a single complex reasoning task for each chart image $I \in \mathcal{I}$ through the following process:

- **Step 1 – Elementary Task Selection.** For each chart, select 4–10 elementary task types from both *Information Extraction* and *Reasoning*, forming a continuous reasoning chain in which tasks are logically connected and require integration of visual and numerical information.

- **Step 2 – Elementary QA Design.** For each selected task type, create a question–answer pair $(q_j, a_j)$, forming a set of subtask QA pairs

$$\mathcal{S} = \{s_j = (q_j, p_j, a_j)\}_{j=1}^{m}, \quad 4 \leq m \leq 10,$$

where $p_j$ denotes the previous task(s) that $q_j$ depends on ($p_j = \emptyset$ if there is no dependency).

- **Step 3 – Complex QA Composition.** An LLM combines the sub-questions $\{q_1, \ldots, q_m\}$ into a single complex question $q_{m+1}$, whose answer $a_{m+1}$ is derived by reasoning over or aggregating the answers of all subtasks. Formally, the complete set of QA pairs is

$$\mathcal{S} = \{s_j\}_{j=1}^{m+1}, \quad 4 \leq m \leq 10,$$

where the final task $s_{m+1} = (q_{m+1}, a_{m+1})$ corresponds to the main composed question and its answer.

**Dataset Summary.** Following this procedure, we constructed 200 complex reasoning QA pairs. The resulting sub-tasks consist of 739 *information extraction* and 713 *reasoning* QA pairs, together forming a structured benchmark for multi-step chart reasoning.

### 3.3 QA Verification and Revision

**Multi-Dimensional Scoring.** Each QA pair is independently reviewed by three annotators across three dimensions: (1) *Semantic Alignment* — whether the question clearly and accurately reflects the chart content without ambiguity or external references; (2) *Reasoning Consistency* — whether the sub-questions form a coherent, logically connected reasoning chain; and (3) *Answer Correctness* — whether the reference answer exactly matches the chart data. Each dimension is rated on a 0–5 scale, with higher scores indicating better quality.

**Iterative Refinement.** After scoring, QA pairs with an average score below 4 are revised. Revisions focus on clarifying ambiguous wording, strengthening the logical connections between sub-questions and the main question, and correcting any inaccuracies in answers. All revised QA pairs are re-evaluated to ensure logical continuity and maintain high-quality reasoning in the final dataset.

### 3.4 Image Variants Generation

Finally, for each chart, we generate four additional visual variants to evaluate the model's robustness under varied visual conditions: a *blurred* version using Gaussian smoothing, a *noisy* version with random pixel perturbations, a *watermarked* version with overlayed text distractions, and an *annotation-removed* version where numeric labels are removed while retaining the chart structure. Together with the original chart, these five versions allow systematic evaluation of robustness and reasoning across different visual conditions, using the same QA pairs for all variants. More details image variants can be found in Appendix A.3.

### 3.5 Statistics

Our dataset is systematically designed to cover both diverse chart types and a wide spectrum of reasoning tasks, enabling thorough evaluation of multimodal models' chart understanding capabilities. It includes 20 distinct chart categories with a total of 200 original chart samples, spanning common single-series and multi-series visualizations such as bar charts, line charts, stacked charts, pie charts, and radar charts. For each original chart, one complex reasoning question is manually created and further decomposed into multiple sub-questions, resulting in approximately 1,652 question-answer (QA) pairs, comprising 739 information extraction questions and 713 reasoning questions (see Fig. 3 for detailed distribution).

From the image perspective, each of the 200 original charts is rendered into five visual variants—*Original Chart with Annotations*, *Blurred Variant*, *Noisy Variant*, *Watermarked Variant*, and *Anno-Removed Variant*—yielding a total of 1,000 chart images. Each variant inherits all QA pairs

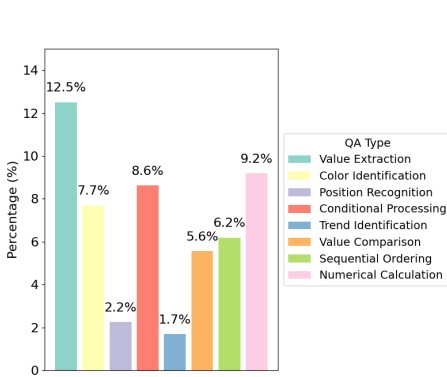

Figure 3: Average proportion of each QA category across complex reasoning QAs.

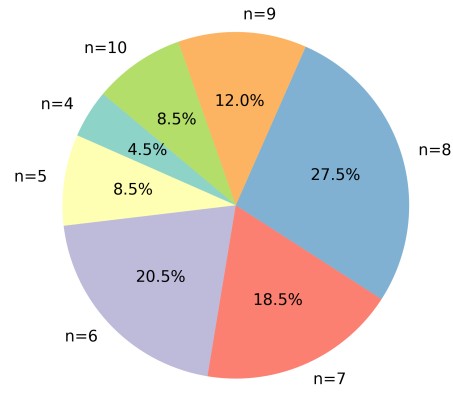

Figure 4: Distribution of complex reasoning questions by number of atomic QAs ($n$ = number of atomic QAs).

from its original chart, resulting in 8,260 image-question-answer (image-QA) pairs across the entire dataset, which significantly enhances the dataset's utility for evaluating model robustness and generalization.

Regarding task design, all questions are grouped into three main types: information extraction, data analysis, and complex reasoning. The information extraction and data analysis tasks are derived from the decomposition of complex reasoning questions. Each complex reasoning question is broken down into 4 to 10 Elementary QAs, with the distribution of atomic QAs counts shown in Fig. 4.

## 4 EVALUATION STRATEGY

We evaluate model performance on our step-wise chart understanding benchmark, considering reasoning correctness, robustness to visual perturbations, and error propagation.

Let the dataset consist of $n$ samples, each with $m_i + 1$ QA pairs:

$$\mathcal{S}_i = \{s_j = (q_j, P_j, a_j^*)\}_{j=1}^{m_i+1},$$

where $q_j$ is the $j$-th question in the $i$-th sample, $P_j = \{q_k\}_{k \in \text{Pre}(j)}$ is the set of prerequisite questions that $q_j$ depends on, $a_j^*$ is the ground-truth answer to $q_j$, and the first $m_i$ questions are sub-tasks while $q_{m_i+1}$ is the main question.

Let $f_\theta$ denote the model under evaluation, which maps a question and a set of prerequisite answers to a predicted answer:

$$f_\theta(\{\hat{a}_k\}_{k \in P_j}, q_j) \to \hat{a}_j,$$

where $\hat{a}_j$ is the model's predicted answer for question $q_j$, and $\hat{a}_k$ denotes the model-predicted answer for a prerequisite question $q_k$.

**Step Accuracy (SA).** Step Accuracy evaluates each question independently, ignoring error propagation from prerequisites. The prediction for $q_j$ uses either the ground-truth answers (*oracle*) or the model-generated answers (*real*) for its prerequisites:

$$\text{SA}_x = \frac{1}{\sum_{i=1}^n m_i + 1} \sum_{i=1}^n \sum_{j=1}^{m_i+1} \text{ACC}\Big(f_\theta(\{a_k^x\}_{k \in P_j}, q_j), a_j^*\Big), \quad x \in \{o, r\},$$

where:

$a_k^o = a_k^*$ (oracle, ground-truth prerequisite answer), $\quad a_k^r = \hat{a}_k$ (real, model-generated prerequisite answer).

**Pipeline Accuracy (PA).** Pipeline Accuracy measures correctness considering error propagation: a question $q_j$ is counted as correct only if its prediction and all its prerequisite answers are correct.

Table 1: Step-level correctness of model predictions and performance under different perturbation settings (SA$_o$ and SA$_p$). The best results in each category are highlighted in bold.

| Model Name | Origin | | Noisy | | Blurred | | Watermarked | | Anno-removed | |
|---|---|---|---|---|---|---|---|---|---|---|
| | SA$o$ | SA$p$ | SA$o$ | SA$p$ | SA$o$ | SA$p$ | SA$o$ | SA$p$ | SA$o$ | SA$p$ |
| *General-Purpose MLLMs* | | | | | | | | | | |
| gemini-2.0-flash | **92.33** | **88.94** | **91.38** | **89.19** | **92.16** | **86.91** | **90.26** | **87.52** | **83.29** | **62.50** |
| gpt-4o | 83.25 | 78.51 | 82.54 | 76.24 | 83.01 | 77.85 | 81.75 | 77.54 | 76.22 | 59.46 |
| Qwen2.5-VL-7B | 82.22 | 77.72 | 81.34 | 75.39 | 82.67 | 76.99 | 81.93 | 75.28 | 75.12 | 54.61 |
| InternVL2_5-8B | 72.96 | 61.56 | 73.82 | 61.28 | 73.75 | 62.47 | 70.62 | 58.97 | 64.79 | 47.98 |
| MiniCPM-o-2_6 | 68.67 | 56.11 | 68.67 | 55.34 | 69.46 | 56.33 | 66.93 | 50.52 | 61.44 | 32.94 |
| deepseek-vl-7b-chat | 49.43 | 32.56 | 48.92 | 32.64 | 48.50 | 32.12 | 47.59 | 31.56 | 48.14 | 31.24 |
| Janus-Pro-7B | 50.33 | 34.18 | 50.24 | 33.15 | 49.56 | 32.78 | 46.65 | 30.04 | 44.22 | 27.64 |
| Phi-4-multimodal | 75.79 | 63.46 | 75.61 | 62.58 | 75.32 | 63.36 | 73.49 | 60.17 | 70.66 | 48.02 |
| InternVL2_5-2B | 52.46 | 40.23 | 52.39 | 41.61 | 51.07 | 38.90 | 51.48 | 37.34 | 47.18 | 26.67 |
| Qwen2.5-VL-3B | 73.99 | 67.33 | 72.16 | 65.15 | 72.33 | 64.03 | 71.34 | 63.71 | 65.33 | 45.35 |
| *Chart-Specific MLLMs* | | | | | | | | | | |
| ChartMoE-8B | 64.90 | 54.56 | 63.27 | 53.97 | 63.10 | 53.45 | 60.66 | 50.87 | 59.35 | 48.46 |
| chartgemma | 47.22 | 30.67 | 46.56 | 31.15 | 46.43 | 30.95 | 42.35 | 24.40 | 42.95 | 21.74 |
| TinyChart-3B | 39.17 | 32.48 | 38.31 | 31.38 | 37.32 | 29.69 | 35.67 | 27.08 | 32.43 | 20.80 |

Formally:

$$\text{PA} = \frac{1}{\sum_{i=1}^{n} m_i + 1} \sum_{i=1}^{n} \sum_{j=1}^{m_i+1} \mathbf{1} \Big[ f_\theta(\{a_k^r\}_{k \in P_j}, q_j) = a_j^* \ \wedge \ \forall q_k \in P_j, \ a_k^r = a_k^* \Big].$$

**Robustness under Visual Perturbations.** To evaluate robustness, the model is tested on perturbed chart variants:

$$\mathcal{V} = \{\text{blurred, noisy, watermarked, annotation-removed}\}.$$

For any metric $M \in \{\text{SA}_o, \text{SA}_r, \text{PA}_o, \text{PA}_r\}$, the Average Performance Degradation (APD) is defined as:

$$\text{APD}_M = \frac{M_{\text{clean}} - \frac{1}{|\mathcal{V}|} \sum_{v \in \mathcal{V}} M_v}{M_{\text{clean}}}.$$

## 5 EXPERIMENTS

### 5.1 BASELINE

We evaluate a comprehensive set of thirteen vision-language models, including ten general-purpose MLLMs (gemini-2.0-Flash Comanici et al. (2025), gpt-4o Hurst et al. (2024), InternVL2.5-2B Chen et al. (2024b), Qwen2.5-VL-3B-Instruct Bai et al. (2025), Phi-4-Multimodal-Instruct Abouelenin et al. (2025), DeepSeek-VL-7B-Chat Lu et al. (2024), Qwen2.5-VL-7B-Instruct Bai et al. (2025), Janus-Pro-7B Chen et al. (2025), InternVL2.5-8B Chen et al. (2024b), and MiniCPM-o-2.6 Yao et al. (2024), and three chart-specialized MLLMs (ChartGemma Masry et al. (2024), TinyChart-3B Zhang et al. (2024), and ChartMoE-8B Xu et al. (2024). All models are evaluated using their default configurations, ensuring a fair comparison across general-purpose and domain-specific approaches.

### 5.2 MAIN RESULTS

Tables 1 and 2 report the quantitative results across clean charts and perturbed variants. Several important observations can be drawn:

**Step-level vs. pipeline reasoning.** On clean charts, most models achieve relatively high **Step Accuracy (SA$o$)**, indicating that they can often answer individual sub-questions correctly when provided

Table 2: **Main Results.** We evaluate models on their ability to complete reasoning chains without error propagation, where a prediction is considered correct only if both the final answer and all of its direct and indirect dependencies are corrct (PA*o*/PA*p*) under different perturbation settings.

| Model Name | Origin | | Noisy | | Blurred | | Watermarked | | Anno-removed | |
|---|---|---|---|---|---|---|---|---|---|---|
| | ACC | PA | ACC | PA | ACC | PA | ACC | PA | ACC | PA |
| *General-Purpose MLLMs* | | | | | | | | | | |
| **gemini-2.0-flash** | 84.0 | 54.5 | 83.0 | 58.0 | 82.0 | 58.0 | 87.0 | 54.5 | 55.0 | 28.0 |
| gpt-4o | 72.0 | 44.5 | 70.5 | 43.0 | 71.5 | 42.0 | 71.0 | 41.0 | 45.5 | 21.0 |
| Qwen2.5-VL-7B | 56.0 | 32.0 | 54.0 | 30.0 | 59.0 | 31.0 | 56.5 | 29.5 | 42.5 | 15.5 |
| InternVL2_5-8B | 53.0 | 30.5 | 52.5 | 28.5 | 52.0 | 29.0 | 50.0 | 26.0 | 40.5 | 9.5 |
| MiniCPM-o-2_6 | 42.5 | 6.5 | 45.0 | 8.5 | 41.0 | 8.0 | 34.5 | 4.5 | 20.5 | 0.0 |
| deepseek-vl-7b-chat | 9.5 | 1.0 | 11.0 | 0.5 | 10.5 | 0.5 | 9.5 | 0.0 | 7.5 | 0.0 |
| Janus-Pro-7B | 10.0 | 1.5 | 9.5 | 2.5 | 8.5 | 2.5 | 8.5 | 1.5 | 9.0 | 0.0 |
| Phi-4-multimodal | 35.5 | 7.5 | 28.5 | 6.0 | 46.0 | 10.5 | 36.0 | 5.0 | 25.5 | 5.0 |
| InternVL2_5-2B | 22.0 | 0.0 | 23.5 | 1.0 | 20.5 | 0.5 | 22.0 | 0.5 | 18.0 | 0.0 |
| Qwen2.5-VL-3B | 48.5 | 30.5 | 46.5 | 29.5 | 49.0 | 29.0 | 45.0 | 27.5 | 22.0 | 11.5 |
| *Chart-Specific MLLMs* | | | | | | | | | | |
| ChartMoE-8B | 40.5 | 6.0 | 41.0 | 7.0 | 39.5 | 6.0 | 38.50 | 6.5 | 21.5 | 0.0 |
| chartgemma | 8.5 | 0.0 | 13.5 | 0.0 | 12.5 | 0.0 | 10.5 | 0.0 | 10.5 | 0.0 |
| TinyChart-3B | 10.0 | 0.5 | 10.0 | 0.5 | 11.0 | 0.0 | 12.0 | 0.5 | 8.0 | 0.0 |

with oracle prerequisites. However, **Pipeline Accuracy (PA*o*/PA*p*)** is consistently lower, as it requires models to maintain correctness across the entire reasoning chain. This large gap suggests that although many models can solve isolated reasoning steps, they struggle to preserve logical consistency when multiple dependencies are involved.

**Reasoning path consistency.** The contrast between Step Accuracy (SA) and Pipeline Accuracy (PA) shows that obtaining a correct final answer does not imply following a valid reasoning trajectory. For instance, GPT-4o reaches 83.25% SAo but only 72.0% PAo, meaning some correct outputs arise despite flawed intermediate steps. Such discrepancies suggest that models often rely on shortcuts or error cancellation rather than faithful reasoning. This reinforces the need to evaluate intermediate steps: without path consistency, final-answer accuracy alone risks overestimating a model's true reasoning ability and obscuring its reliability in real-world analytical tasks.

**Error propagation in pipeline evaluation affects multi-step reasoning.** Comparing Oracle and pipeline metrics shows that inaccuracies in earlier reasoning steps can negatively impact subsequent questions. This illustrates that even when final answers might be correct in isolation, the dependency between sub-questions can propagate errors along the reasoning chain.

**General-purpose vs. chart-specific models.** Interestingly, chart-specialized models do not consistently outperform general-purpose MLLMs. While models like ChartMoE-8B capture some chart-specific cues, their overall reasoning performance remains lower than general-purpose systems such as Gemini-2.0-Flash and GPT-4o. This suggests that large-scale multimodal pretraining with diverse reasoning signals provides stronger generalization than domain-focused fine-tuning alone. However, specialized models may still hold potential if combined with advanced reasoning strategies or error-aware mechanisms.

## 5.3 ROBUSTNESS TO VISUAL PERTURBATIONS

As shown in Table 3, models exhibit differing levels of robustness under four perturbation types: noise, blur, watermarking, and annotation removal. Overall, performance is relatively stable under blur and noise, with only moderate decreases in both SA and PA scores, suggesting that models can still rely on global structural cues when local fidelity is degraded. In contrast, annotation removal and watermarking result in the most pronounced accuracy drops. The robustness metrics DASA and DAPA further quantify these effects, highlighting that information integrity—particularly the presence of textual annotations—is essential for maintaining consistent multi-step reasoning. Taken

Table 3: Model robustness under different metrics. The best results for each category is highlighted in **bold**, and the second-best is indicated with underline.

| Model Name | APD$_{\text{SA}o}$ | APD$_{\text{SA}p}$ | APD$_{\text{PA}o}$ | APD$_{\text{PA}p}$ |
|---|---|---|---|---|
| gemini-2.0-flash | 3.31 | 8.33 | 6.56 | 8.68 |
| GPT-4o | 2.85 | 7.31 | 5.47 | 7.96 |
| Qwen2.5-VL-7B | 2.38 | 9.20 | 3.27 | 10.87 |
| InternVL2_5-8B | 3.04 | 6.31 | 4.68 | 5.91 |
| MiniCPM-o-2_6 | 2.98 | 13.06 | 7.50 | 13.94 |
| deepseek-vl-7B | **2.31** | **2.06** | **2.56** | **5.53** |
| Janus-Pro-7B | 5.29 | 9.59 | 10.47 | 8.74 |
| Phi-4-multimodal | 2.67 | 7.76 | 5.87 | 9.17 |
| InternVL2_5-2B | 3.68 | 10.19 | 8.38 | 12.10 |
| Qwen2.5-VL-3B | 5.00 | 11.54 | 8.22 | 14.63 |
| ChartMoe | 5.09 | 5.27 | 9.59 | 7.16 |
| chartgemma | 5.61 | 11.77 | 10.36 | 17.03 |
| TinyChart-3B | 8.27 | 16.14 | 13.28 | 18.48 |

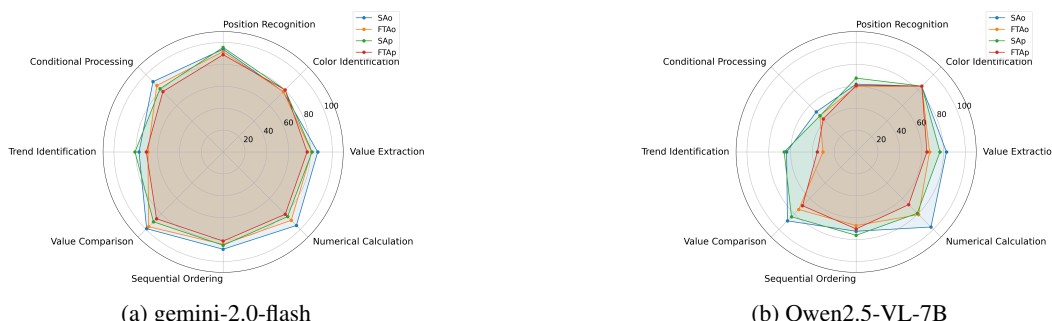

(a) gemini-2.0-flash        (b) Qwen2.5-VL-7B

Figure 5: Model Performance on Each Task Category (Left: gemini-2.0-flash, Right: Qwen2.5-VL-7B).

together, annotation removal and watermarking exert the strongest negative impact, noise has a moderate effect, and blur is the least disruptive.

## 5.4 TASK-LEVEL PERFORMANCE COMPARISON

Figure 5 compares Gemini-2.0-Flash and Qwen2.5-VL-7B across elementary task types, showing stronger and more consistent performance by Gemini in extraction, calculation, and ordering tasks, while Qwen exhibits notable drops in conditional and color-based reasoning. The SAo-to-FTAo decline in both models highlights error propagation, especially in Qwen, reinforcing the need for step-level evaluation.

## 6 CONCLUSION

This paper introduces a fine-grained benchmark for evaluating MLLMs' chart understanding capabilities through decomposed reasoning chains and visually perturbed variants. By assessing both intermediate steps and final answers, we reveal that while models like Gemini-2.0-Flash perform strongly on isolated sub-tasks, maintaining logical consistency across multi-step reasoning remains a challenge. Our findings underscore the limitations of final-answer-only metrics and highlight the importance of robustness to visual perturbations.

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

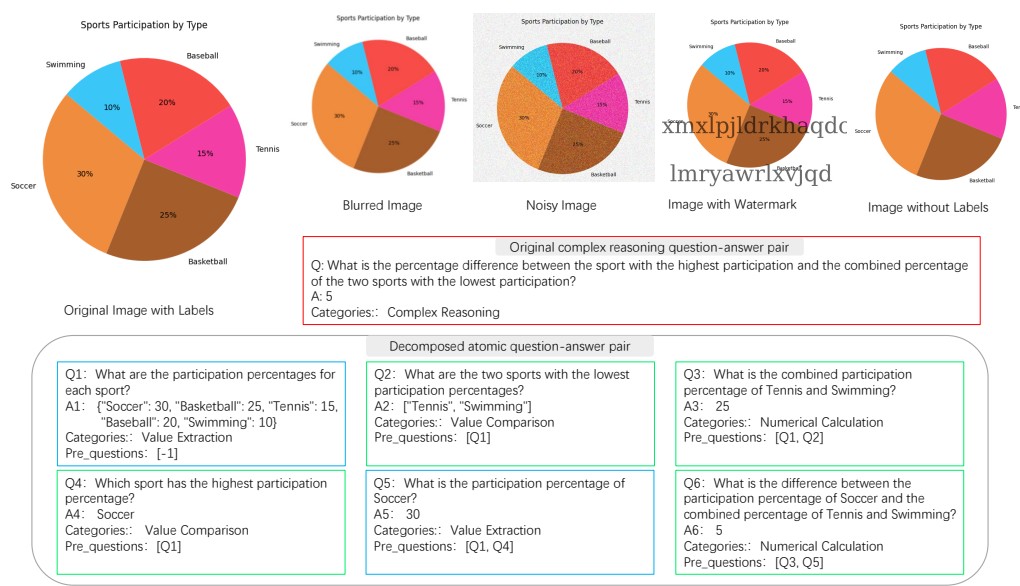

Figure 6: The benchmark example in our dataset

# A APPENDIX

## A.1 ADDITIONAL DATA CONSTRUCTION DETAILS

We provide additional visualizations and more detailed data construction descriptions.

## A.2 ILLUSTRATIVE EXAMPLE OF COMPLEX REASONING TASK GENERATION

*Atomic QAs:* Consider a chart showing the percentage contribution of different renewable energy sources. The selected atomic tasks might be: Value Extraction, Distribution Identification, and Numerical Calculation. The corresponding sub-task QA pairs could be:

- *Value Extraction:* "What is the 'Percentage Contribution' for each 'Energy Source'?" Answer: Solar 35, Wind 25, Hydro 20, Biomass 15, Geothermal 5.

- *Distribution Identification:* "What are all the 'Energy Source' in the range of 'Percentage Contribution' from 23 to 38?" Answer: Solar, Wind.

- *Numerical Calculation:* "What is the sum of the 'Percentage Contribution' for Solar and Wind?" Answer: 60.

- *Distribution Identification:* "What are all the 'Energy Source' in the range of 'Percentage Contribution' from 0 to 22?" Answer: Hydro, Biomass, Geothermal.

- *Numerical Calculation:* "What is the sum of the 'Percentage Contribution' for Hydro, Biomass, and Geothermal?" Answer: 40.

- *Numerical Calculation:* "What is the difference between the total percentage contribution of [Solar, Wind] and the total percentage contribution of [Hydro, Biomass, Geothermal]?" Answer: 20.

*Complex reasoning QA:*

> "What is the difference between the total percentage contribution of renewable energy sources contributing between 23% and 38% and the total percentage contribution of sources contributing between 0% and 22%?"

The final answer *A* aggregates the answers from all sub-tasks: *20*.

## A.3 Image Variants Generation

This subsection details the generation of several image variants designed to evaluate the model's robustness under various visual perturbations. Each variant introduces a distinct form of distortion or noise, allowing for a comprehensive assessment of the model's ability to accurately extract relevant information despite changes in image quality or the introduction of irrelevant elements.

- *Blurred Variant:* This variant is generated by applying a Gaussian blur to the image using OpenCV's `cv2.GaussianBlur`, with a default kernel size of 5 and a standard deviation of 0. The blur reduces image sharpness while preserving the overall structure of the chart, aiming to evaluate the model's robustness to visual degradation and its ability to extract information under softened or less distinct visual conditions.

- *Noisy Variant:* This variant is generated by adding random integer noise to the image pixels, with values ranging in $[-50, 50]$, using NumPy. The added noise simulates real-world visual disturbances, aiming to evaluate the model's robustness in extracting key information under realistic noisy conditions.

- *Watermarked Variant:* This variant is created by overlaying randomly generated watermark text onto the image using PIL's `ImageDraw` library. By default, two text strings—each 10 to 20 characters long—are rendered in dark gray and placed at random positions on the image. These watermarks act as irrelevant visual distractions, allowing us to test the model's ability to focus on the main content despite the presence of such noise.

- *Annotation-Removed Variant:* In this variant, numeric annotations such as data labels and value markers are removed from the chart while retaining the original structure, including axes, bars or lines, and overall layout. By omitting explicit numerical information, this version tests the model's ability to interpret the chart based purely on visual cues such as shape, relative position, and trend patterns.

