# OpenReview forum: "Benchmarking Complex Chart Reasoning via Sub-question Decomposition and Variant-based Robustness Analysis"
_ICLR.cc/2026/Conference — ICLR 2026 Conference Withdrawn Submission_

### Official Review · Reviewer_sgkJ · 2025-10-20

**Soundness:** 3
**Presentation:** 2
**Contribution:** 2
**Rating:** 2
**Confidence:** 4

**Summary:**

This paper proposes a benchmark for chart understanding of MLLMs, which contains 20 chart types, 200 seed charts, and 1,652 sub-questions. Experimental results show a performance drop of MLLMs and provide analysis and findings to system design of chart analysis.

**Strengths:**

1. The motivation for the research question is promising and interesting. Different from previous work, this paper investigates MLLMs' capability on chart understanding by interpretable sub-question.
2. The dataset should be an important contribution, facilitating follow-up work.

**Weaknesses:**

1. The size of seed charts is small (N=200), which may impair the reliability of the experimental results. Although the author expanded the number to one thousand charts through data augmentation to study robustness, essentially they are still variants of the 200 charts, as well as the corresponding 1,652 question-answer pairs.
2. The charts in the benchmark dataset are generated rather than being real-world charts, which impairs the meaningfulness and reliability of experimental results for real-world scenarios.
3. The repository that the authors provided in the paper is empty (by 10.20). It is allowed not to provide a repo during the reviewing process. However, providing an empty repository is not a good way that may affect the fairness, as some reviewers may prefer work that claims providing a repository.
4. Line 204, the taxonomy of the proposed tasks needs some justifications. Why were these tasks chosen? Do they comprehensively cover all aspects of chart understanding research? What considerations did the author have in this regard? These are all matters that need to be discussed.
5. Section 3.3, the process of quality control needs more details, e.g., the agreement between annotators, the dynamics of QA pairs (scores and specific content) over iterations.

**Questions:**

My suggestions and questions are placed under "Weaknesses", as they correspond to each weakness.

---

### Official Review · Reviewer_VAV7 · 2025-10-25

**Soundness:** 3
**Presentation:** 2
**Contribution:** 3
**Rating:** 6
**Confidence:** 4

**Summary:**

This paper is about multimodal large language model evaluation. The authors examine intermediate reasoning validity and robustness to visual perturbations as part of the evaluation. Specifically, they introduce a fine-grained benchmark that decomposes each complex question into interpretable sub-questions and tests models under five visual perturbations (blur, noise, watermark, label removal, color distortion). Spanning 20 chart types, 200 base charts yield 1,652 sub-questions and 8,260 QA pairs across 1,000 images. They show current MLLMs may fail in the evaluations.

**Strengths:**

1. The evaluation of potential distribution shifts and the validity of the reasoning process is worth exploring.
2. The findings of this paper highlight the necessity of process-oriented diagnostics to ensure trustworthy deployment in real-world, low-quality visual environments.
3. The authors have provided detailed results of the evaluation.

**Weaknesses:**

1. The paper stops at observations and there is no methodology for mitigating the problem discovered by the authors. While the observations and findings about chart interpretation under visual corruptions and the validity of intermediate reasoning steps are valuable, this paper lacks a solution for mitigating the problem.
2. This paper evaluates the models' performance under distribution shifts. Similar issues have been studied by prior works [1-2]. The authors should include them with additional discussions.
3. The authors should discuss how chart interpretation evaluation is different from general multimodal evaluations [3-4].

[1] Understanding multimodal llms under distribution shifts: An information-theoretic approach

[2] Multifaceted Evaluation of Audio-Visual Capability for MLLMs: Effectiveness, Efficiency, Generalizability and Robustness

[3] Evaluating mllms with multimodal multi-image reasoning benchmark

[4] MMEvalPro: Calibrating Multimodal Benchmarks Towards Trustworthy and Efficient Evaluation

**Questions:**

Please refer to the weakness section.

---

### Official Review · Reviewer_SMdW · 2025-10-30

**Soundness:** 2
**Presentation:** 2
**Contribution:** 2
**Rating:** 2
**Confidence:** 4

**Summary:**

The paper introduces CHART-FGR, a benchmark for chart QA that (1) decomposes each complex question into a chain of atomic sub-questions to evaluate step-wise reasoning and pipeline consistency, and (2) renders five visual variants per chart to test robustness (clean, blur, noise, watermark, and annotation/label removal). The dataset spans 20 chart types, 200 base charts, ~1,652 sub-questions and 8,260 QA pairs across 1,000 images. Metrics include Step Accuracy with oracle vs real prerequisites (SAo/SAp) and Pipeline Accuracy (PAo/PAp). Results across general-purpose and chart-specialized MLLMs show large SA→PA gaps and sharp drops under annotation removal.

**Strengths:**

- The setting that combines step-wise/pipeline accuracy and perturbed images in chart understanding is novel, and authors conducted a thorough evaluation how these variables {step, pipeline acc} x {clean, perturbed charts} lead to different performance for different models.
- Synthetic data generation and manual inspection and filtering make the study more controlled and rigorous.

**Weaknesses:**

- The contribution feels incremental. Prior work already studies the effect of visual perturbations on chart QA (e.g., CHAOS: Chart Analysis with Outlier Samples), and there is also existing work on principled reasoning decomposition (e.g., GRS-QA for graph reasoning, although not chart-specific). Visual perturbation and reasoning decomposition seem orthogonal; applying both to chart understanding can yield domain-specific observations, but it is not yet clear that this combination provides fundamental insights for the broader community. A stronger connection to prior work, along with evidence showing corroboration or contradiction, would help clarify the novelty and value.
- The pipeline for generating sub-questions and chart variants appears largely automatable (aside from human quality checks). Are there methods to automate the quality verification step as well? If so, an interesting extension would be to train models on automatically generated data and analyze whether errors arise from data distribution issues versus limitations in model architecture or training paradigms.
- While manual filtering based on visual readability and data rationality ensures higher-quality charts, this process may introduce systematic biases. For instance, chart styles less prone to occlusion or outliers may be over-represented. A careful analysis demonstrating that the filtering removes malformed cases without skewing chart style or complexity distribution would be helpful in strengthening the dataset validity.
- The study evaluates a wide range of models, but it would be particularly informative to analyze scaling effects within model families (e.g., Qwen 2.5 VL small vs larger variants). Understanding how model size impacts reasoning consistency and robustness to perturbations would provide deeper insight into capacity-related behavior.

**Questions:**

- Are the labels between L383-384 in table 2 PA_o and PA_p instead of ACC and PA? I assume they mean the same thing but it would be good to have consistent labels.
- For each complex item, do you run the model in multi-turn mode where the next sub-Q conditions on the previous predicted answer (real prerequisites), or do you evaluate each sub-Q independently and only aggregate for PA? The definitions suggest both oracle and real modes exist, but the run-time protocol isn’t fully explicit.
- Code from the anonymous github appears to be empty — could authors fix it?

---

### Official Review · Reviewer_TUSr · 2025-10-31

**Soundness:** 3
**Presentation:** 3
**Contribution:** 2
**Rating:** 2
**Confidence:** 4

**Summary:**

This paper introduces a fine-grained benchmark for complex chart reasoning that combines sub-question decomposition with variant-based robustness analysis. Each complex question is systematically broken down into logically coherent sub-questions, enabling interpretable evaluation of reasoning pathways and precise error diagnosis. Additionally, the author constructs five variants of each chart, including blurred, noise-added, watermark-added, and label-removed, to measure model resilience under degraded conditions.

**Strengths:**

- The idea of the benchmark is interesting and important for assessing the intersection of MLLMs and structured data understanding.
- The paper is well-written and easy to follow, with a clear structure.
- The motivation behind the work is interesting and well-presented.

**Weaknesses:**

- The annotator or reviewer pool is unclear. The benchmark is heavily filtered and annotated by humans. A discussion and statistics should be provided to warrant the diversity of the annotators and address concerns of human bias if the reviewer pool is small. Questions such as: How many reviewers are in the pool? What are the instructions for the human evaluators?

- Potential bias of generated data from LLM. The initial JSON data for charts are generated by an LLM. However, synthetic data tends to carry the bias from the LLM; for example, it may have monotonic trends, like only upward or downward. Was this issue observed in the chart domain data? How was this evaluated? Can the author analyze the pattern diversity of the proposed benchmark?

- Models in the comparison are outdated or limited. The models compared in Tables 1 and 2 are either old closed-source models or open-source models at a similar scale. What is the performance of larger models? Does the trend persist? For example, what is the performance of Gemini Pro and GPT-5? Testing on 1,000 samples should be feasible. Or, larger open-source models should be studied.for example, QWen 2.5VL 32B and 72B.

- The analysis findings are interesting; however, they lack in-depth findings. If the step-level performance is not 100%, then Pipeline accuracy will drop quadratically, which is general and obvious behavior. A more in-depth analysis combining the pipeline accuracy should be jointly provided to reflect meaningful findings for the community. For example, what is the percentage of incorrect answers in each pipeline questions set? What is the distribution of question types that models tend to fail on? An LVLM can achieve high step-level accuracy but low pipeline accuracy, even if it achieves good general performance, if some essential capability for chart understanding is missing. Without in-depth analysis, the benefit that the paper can bring to the community is limited.

**Questions:**

- What is the human step-level performance? Under the blur effect, I believe even humans can make mistakes as some details may be hard to read. Gemini-2.0-flash reaches 90%+ accuracy on the step-level questions; doesn't this indicate that the step-level questions are saturated?

- When performing inference, are answers for each sub-question generated separately, or is it more like a multi-turn conversation? From the equation in line 311, it seems that only model-predicted answers for prerequisite questions are provided, not the previous sub-questions. What is the design purpose behind this, instead of using a multi-turn conversation?

---

### Note · Authors · 2025-11-12

I have read and agree with the venue's withdrawal policy on behalf of myself and my co-authors.